# Adversarial Attacks through Value-Guided Transition Modeling in Deep Reinforcement Learning

Thomas O'Cuilleanain[1], Juan Cardenas-Cartagena[1], and Matthia Sabatelli[*1]

[1]University of Groningen
[1]m.sabatelli@rug.nl

## Abstract

Efficient adversarial attacks on deep reinforcement learning agents rely on identifying critical states. Prior work uses learned transition models with environment-specific metrics to predict and lure the victim agent to such states. We propose a value-guided attack that integrates the victim policy's value function as an environment-agnostic metric into both transition model training and state evaluation. From our preliminary results in the `Pong` environment from the Arcade Learning Environment, our method achieves comparable performance degradation to prior work while requiring roughly half as many attacks.

## 1 Background

**Reinforcement Learning:** Reinforcement Learning (RL) environments are modeled as Markov Decision Processes (MDP) [1], defined by

$$\mathcal{M} = (\mathcal{S}, \mathcal{A}, p, \mathcal{R}, \gamma),$$

where $\mathcal{S}$ is the state space, $\mathcal{A}$ the action space, $p(s_{t+1} \mid s_t, a_t)$ the transition function, $\mathcal{R}(s_t, a_t, s_{t+1})$ the reward function, and $\gamma \in [0, 1]$ the discount factor. At each time-step $t$, the agent observes the current state $s_t \in \mathcal{S}$, selects an action $a_t \in \mathcal{A}$ according to its policy $\pi$, transitions to a next state $s_{t+1} \sim p(\cdot \mid s_t, a_t)$ and receives a scalar reward $r_t = \mathcal{R}(s_t, a_t, s_{t+1})$. The agent's behavior is governed by its policy $\pi(a_t \mid s_t)$, which defines a probability distribution over actions given the current state. The state-value function represents the expected cumulative discounted return when starting from $s_t$ and following $\pi$ thereafter:

$$V^\pi(s) = \mathbb{E}_\pi \left[ \sum_{k=0}^{\infty} \gamma^k r_{t+k} \mid s_t = s \right]. \quad (1)$$

In this work, we leverage the value function, $V^\pi(s)$, as an environment-agnostic metric for identifying critical states to which we lure the victim policy.

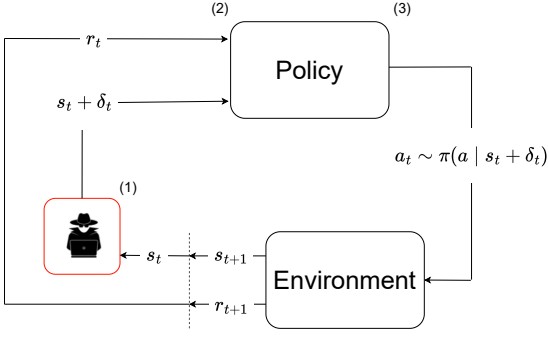

**Figure 1.** General schema of test-time, state-based adversarial attacks in DRL: (1) the adversary intercepts the true observation and injects a perturbation $\delta_t$ into the state $s_t$ (2) which is passed to the agent. (3) The agent then samples an action $a_t \sim \pi(a \mid s_t + \delta_t)$ which is executed in the environment.

**Adversarial Attacks:** Test-time adversarial attacks on Deep Reinforcement Learning (DRL) agents aim to manipulate the behavior of a trained victim policy to achieve an ulterior objective (e.g. performance degradation). These attacks exploit the sensitivity of trained policies by perturbing their input in order to induce the policy into taking (targeted) suboptimal actions during evaluation. We denote the perturbation added to $s_t$ as $\delta_t$. Figure 1 illustrates this framework.

Sun et al. [2] propose Critical Point Attack (CPA) which identifies critical states to make adversarial attacks more efficient by learning a parametrized transition model[1] $f_\theta : \mathcal{S} \times \mathcal{A} \to S$ of the environment, where $\theta$ are the parameters. This model follows the same architecture proposed by Oh et al. [3]. Given a dataset of $N$ collected trajectories from the victim agent $D = \left\{ \left( (s_0^{(i)}, a_0^{(i)}), \ldots, (s_{T_i}^{(i)}, a_{T_i}^{(i)}) \right) \right\}_{i=0}^{N}$, and a prediction horizon with length $K$, $\theta$ is optimised using the loss,

$$\text{SE}^{(i)}(t; \theta) = \left\| \hat{s}_t^{(i)}(\theta) - s_t^{(i)} \right\|_2^2, \quad (2)$$

$$L_{\text{CPA}}(\theta) = \frac{1}{2K} \sum_{i,t} \sum_{k=1}^{K} \text{SE}^{(i)}(t + k; \theta), \quad (3)$$

---

[*]Corresponding Author.

[1]Here "transition model" denotes a learned environment dynamics predictor, not the MDP transition function.

where $\hat{s}_t^{(i)}(\theta) := f_\theta(\hat{s}_{t-1,\theta}^{(i)}, a_{t-1}^{(i)})$, $t > 0$. For simplicity, we omit $\theta$ in the $\hat{s}$ notation unless required explicitly. Using $f_\theta$, given a rollout horizon length $M$, and the victim policy $\pi$, the adversary predicts the baseline state $\hat{s}_{t+M}^\pi$ starting from $s_t$ by following the victim policy on predicted next states recursively. Next, given an attack horizon of length $K \leq M$, the adversary predicts all possible subsequent states by enumerating all possible action sequences of length $K$. We denote a specific sequence as $\mathbf{a}_{t:t+K} \triangleq (a_i)_{i=t}^{t+K} \in \mathcal{A}^K$, where $\mathcal{A}^K$ is the Cartesian product of $\mathcal{A}$ with itself $K$ times. If $M > K$, then for each such predicted state the adversary continues the rollout using the victim policy for the remaining $M-K$ steps. This results in $|\mathcal{A}^K|$ predicted final states. Using an environment specific divergence function $T : \mathcal{S} \to \mathbb{R}$, the adversary finds the final action sequence from the starting state which maximises the Danger Awareness Metric,

$$\mathrm{DAM}_T(\mathbf{a}_{t:t+K}) = \left| T(\hat{s}_{t+M}^{\mathbf{a}_{t:t+K}}) - T(s_{t+M}^\pi) \right|, \quad (4)$$

$$\mathbf{a}_{t:t+K}^* = \underset{\mathbf{a}_{t:t+K} \in \mathcal{A}^K}{\arg\max} \mathrm{DAM}_T(\mathbf{a}_{t:t+K}), \quad (5)$$

where $\hat{s}_{t+M}^{\mathbf{a}_{t:t+K}}$ denotes the predicted state $\hat{s}$ at time step $t+M$ following action sequence $\mathbf{a}_{t:t+K}$ for $K$-steps and the victim policy $\pi$ for $M-K$-steps thereafter. If, for any given final state, this metric surpasses a threshold $\Delta > 0$, the victim policy is fooled into following the associated action sequence by adding carefully crafted perturbations. These perturbations are computed using the Carlini & Wagner (C&W) attack [4]. The authors state that it is necessary for $T$ to have an environment-specific definition to accurately reflect the potential danger associated with a predicted state. In the `Pong` and `Breakout` environments from the Arcade Learning Environment [5], the authors turn to predicting the RAM state representation of subsequent states. Given the RAM state $s$, the authors define $T(s) = d(s) \cdot p(s)$, where $d : \mathcal{S} \to \mathbb{R}$ is the Euclidean distance between the ball and the paddle and $p : \mathcal{S} \to \{0, 1\}$ is equal to 1 if the ball has been dropped and 0 otherwise.

## 2 Our Method

**Rationale:** The proposed attack employs an environment-agnostic $T$ function based on the state-value function $V^\pi$. In this context, the adversary finds the action sequence $\mathbf{a}_{t:t+K}^*$ as follows,

$$\mathbf{a}_{t:t+K}^* = \underset{\mathbf{a}_{t:t+K} \in \mathcal{A}^K}{\arg\max} \mathrm{DAM}_{V^\pi}(\mathbf{a}_{t:t+K}). \quad (6)$$

From our early experiments using $L_{CPA}$ to optimize $\theta$, we observed that in the `Pong` environment, the final reconstruction loss was minimal, yet the difference in state-value estimates between genuine and predicted states remained large, ultimately hindering the effectiveness of the attack. We hypothesize

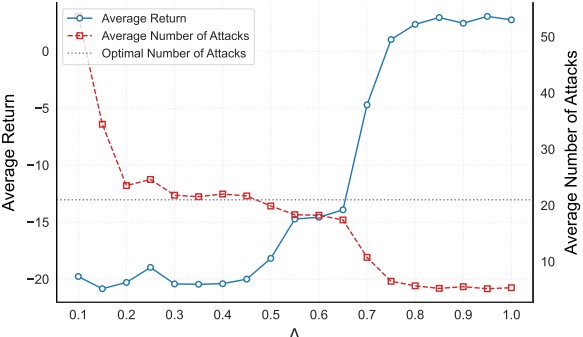

**Figure 2.** Average return and number of attacks vs $\Delta$ against trained A2C victim policy using value-guided attack with $K = M = 1$ in `Pong` averaged over 100 episodes.

that this occurs because, although the reconstruction error is low, the ball in these environments occupies only a few pixels in the observation space. Consequently, the reconstruction loss provides a very weak learning signal for its precise position. These inaccuracies in the ball's position lead to substantial discrepancies in the policy's value estimates, which are highly sensitive to ball position. Therefore, we propose a new loss to optimize $\theta$,

$$\mathrm{SE}_{V^\pi}^{(i)}(t; \theta) = \left\| V^\pi(\hat{s}_t^{(i)}(\theta)) - V^\pi(s_t^{(i)}) \right\|_2^2, \quad (7)$$

$$L_V(\theta) = \frac{1}{2K} \sum_{i,t} \sum_{k=1}^K \mathrm{SE}_{V^\pi}^{(i)}(t+k; \theta), \quad (8)$$

$$L(\theta) = \alpha L_{\mathrm{CPA}}(\theta) + (1-\alpha) L_V(\theta), \quad (9)$$

where $\alpha \in [0, 1]$ allows tuning for different tasks. This guides the reconstruction to produce subsequent states that are not only visually accurate but aligned with $V^\pi$, thereby emphasizing task-relevant features, such as the position of the ball.

**Preliminary Findings:** In our experiments, we optimize $\theta$ using $L$ with $\alpha = 0.1$. We evaluate this attack against a trained victim A2C agent in the `Pong` environment. The lowest return of -21 is achieved after a minimum of 21 attacks, as the agent must drop the ball 21 times. Averaging over 100 episodes, with $K = M = 1$, and threshold $\Delta = 0.35$ the attack achieves an average return of -20.45 in 21.55 attacks (see Figure 2). Compared to the results in Sun et al. [2], **the attack achieves comparable performance degradation in around half the number of attacks ($K = M = 2$).** We plan to test our environment-agnostic adversarial attack in environments in which constructing an environment-specific $T$ function is infeasible; yet estimating the victim policy's value function remains tractable for discrete action spaces.

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
