# OpenReview forum: "Adversarial Attacks through Value-Guided Transition Modeling in Deep Reinforcement Learning"
_NLDL.org/2026/Abstracts_Track — NLDL 2026 Abstracts_

### Official Review · Reviewer_PF8s · 2025-10-24

**Soundness:** 4
**Correctness:** 4
**Rating:** 5
**Confidence:** 3

**Summary:**

This work extends a method for adversarially attacking a (reinforcement learning) victim agent into critical states where the agent's performance degrades. They propose finding these critical states in an environment-agnostic manner by considering an adversarial action sequence based on the value function rather than the environment-specific divergence function. Additionally, they propose introducing a loss term based on the value function when training the transition model.

**Strengths:**

This work is conceptually clean and the choices are well-motivated from either a practical or experimental perspective. The formulation of adversarial attacks in deep reinforcement learning and the usage of the state-value function for guiding the adversarial attack is sound.

**Weaknesses:**

1. What are the assumptions of the action space? I guess the approach only works for discrete action spaces as you rely on the Cartesian product of $\mathcal{A}$ for finding the adversarial action sequence?
2. You introduce a hyperparameter $\alpha$. Is this hard to optimize and dependent on the specific environment?
3. You state that $\hat{s}_{t}^{(1)}(\theta)$ is the output of $f_{\theta}(\hat{s}_{t-1}^{(i)}, a_{t-1}^{(i)})$ but what is $\hat{s}_0^{(i)}$ ?
4. Including results of the environment-specific divergence function $T$ would be nice for comparison in Figure 2.

---

### Official Review · Reviewer_Z2UP · 2025-10-26

**Soundness:** 4
**Correctness:** 3
**Rating:** 4
**Confidence:** 4

**Summary:**

This paper proposes a value guided adversarial attack method on deep reinforcement learning agents. Authors demonstrated that the method achieves comparable degradation in policy performance to CPA, but with roughly half the number of attacks.

**Strengths:**

1. Introduces a new environment-agnostic attack mechanism.
2. The rationale for using the value function is well-justified.
3. Empirical evidence shows the approach achieves comparable damage with approximately half the number of attacks.

**Weaknesses:**

1. Only evaluated on one environment and with a single algorithm.
2. Need exploration of hyper parameter sensitivity or robustness across random seeds.
3. Performance degradation is reported but will be nice some experiments and further discussion of attack stealthiness, perturbation magnitude and transferability.
4. Ablation study to identify and analyze why value alignment improve critical state prediction.
5. Experimental details such as computational cost.

These are just recommendations for a future extended version of this work :)

Good luck!

---

### Official Review · Reviewer_EFEz · 2025-11-02

**Soundness:** 4
**Correctness:** 4
**Rating:** 4
**Confidence:** 2

**Summary:**

The authors propose an environment-agnostic way of performing adversarial attacks on a reinforcement learning (RL) agent by using the agent’s own value function to guide where and how to attack.

They define their adversarial model’s training loss such that predicted future states are not only visually accurate but also yield similar value estimates according to the victim agent’s policy.

Early tests on the Pong game show that their method can degrade performance about as much as previous attacks while requiring roughly half as many attacks.

**Strengths:**

- The property for the attacks to be environment agnostic good for generalization.
- Well-formalized math in general, and the text is clearly written.

**Weaknesses:**

- No visual interpretation of attacks, would be interesting to see $s_t$, along with $\delta_t$.
- The text leans too heavily on formulas without clear intuition for why they were implemented.
- Could use some discussion on how likely the method is to scale past simple environments like arcade-games

---

### Decision · Program_Chairs · 2025-11-05

**Decision:**

Accept

**Comment:**

The abstract is of interest to the community and should be presented at the conference.